# Fractional order ATR-FTIR differential spectroscopy for detection of weak bands and assessing the radiation modifications in gamma sterilized UHMWPE

**Muhammad Mudassir Saeed, Malik Sajjad Mehmood, Muhammad Muddassar** [ID] *

Department of Basic Sciences, University of Engineering and Technology, Taxila, Pakistan

* malik.muddassar@gmail.com

**Data Availability Statement:** All relevant data are within the manuscript.

**Funding:** The author(s) received no specific funding for this work.

## Abstract

This study presents a new method for identifying radiation modifications in UHMWPE polymer samples. The method involves using a mathematical technique called fractional order differential transformation on IR spectra obtained through ATR-FTIR spectroscopy. This new method was compared to existing techniques such as FTIR, XRD, and DSC, and it was found to be more sensitive and accurate in detecting radiation-induced changes in the polymer. The study focused on identifying changes in weak IR bands in the UHMWPE samples caused by gamma sterilization while simulating IR spectra using different orders of fractional derivatives and compared them to experimental spectra. It was found that applying a lower order of differentiation was more suitable for identifying radiation-induced changes in the UHMWPE samples. Using this method, they were able to identify specific changes in the gamma irradiated structure, such as the splitting of a single absorption peak into a doublet, which was only present in the 50 kGy irradiated sample. The study also used correlation index analysis, principal component analysis, and hierarchy cluster analysis to analyze the simulated and experimental spectra. These techniques allowed to confirm the effectiveness of the fractional order differential transformation method and to identify the specific regions of the IR spectra that were affected by radiation-induced changes in the UHMWPE samples. Overall, this study presents a new method for identifying radiation-induced changes in UHMWPE polymer samples that is more sensitive and accurate than existing techniques. By identifying these changes, researchers can better understand the effects of gamma sterilization on medical equipment and potentially develop new methods for sterilization that do not damage the equipment.

## 1. Introduction

Polymer and polymer-based products are becoming increasingly popular due to their low cost, ease of manufacturing, and physical and chemical properties [1–4]. Among the various types of polymers, ultra-high molecular weight polyethylene (UHMWPE) with a molecular weight

**Competing interests:** The authors have declared that no competing interests exist.

of approximately 3–6 million g/mol is widely used in various applications such as electrical insulation, artificial implants, and microelectronics [5, 6]. The properties of UHMWPE can be modified by crosslinking through different chemical and physical methods such as irradiation, organo-silane, and peroxides [5–7].

To analyze UHMWPE for these applications and modifications, various characterization methods such as FTIR, XRD, and DSC are used [7]. However, these methods have limitations in identifying weak bands and small amounts of contaminants, leading to inaccurate and uncertain results. FTIR spectroscopy, in particular, is commonly used for post-modification structural analysis, but it requires spectral pre-processing to enhance sensitivity and accuracy. Unfortunately, FTIR spectroscopy has drawbacks, such as the inability to identify weaker bands and extremely tiny amounts of contaminants, and the findings are sometimes tough to understand and uncertain. Usually this is becoming as it uses data from the sample transmitted/reflected that is detected by the built-in detection system for further analysis. This data might contain linear or quasi linear noise and baseline shift etc. that might responsible for inaccurate assessment of radiation induced modification in UHMWPE. So spectral pre-processing is necessary for accurate assessment, and in this regard, application of differential filters more particularly fractional order differential filters need to be considered for spectral pre-processing and enhancing the sensitivity of assessment protocol and identification of weak IR bands. Identifying weak IR bands and radiation modifications in UHMWPE is crucial for improving the performance of polymer-based products and ensuring their safety in various applications. This study offers a more precise and sensitive approach to UHMWPE characterization, addressing the limitations of current methods and providing valuable insights for future research in this field.

This study aims to address the limitations of current approaches to UHMWPE characterization by analyzing ATR-FTIR spectra of pristine and gamma irradiated samples after applying fractional order differential filters of various orders ranging from 0 to 1. The manuscript provides an overview of fractional order derivatives and describes the methodology of obtaining simulated spectra from ATR-FTIR experimental data. The most sensitive and lowest possible transformation is determined using correlation index analysis and Principal Component Analysis (PCA). Hierarchy cluster analysis (HCA) is then performed using experimental spectra and simulated fractional derivative (FD) spectra to assess radiation modification.

## 2. Background literature

In 1695 L'Hospital asked Leibniz what meaning could be given to the sign $\frac{d^n y}{dx^n}$ after $n = 1/2$. In September 30, 1695, Leibniz replied, "It will lead to confusion, with good results in one day." This discussion leads to a new mathematical branch on the output and integration of the opposite order is also known as Fractional Calculus. Lacroix (1819) was the first to describe the discovery of the order of contradiction in which he write down the following formula [8, 9]

$$\frac{d^{1/2}}{dx^{1/2}}(x^a) = \frac{\Gamma(a+1)}{\Gamma\left(a+\frac{1}{2}\right)} x^{\left(a-\frac{1}{2}\right)}. \tag{1}$$

$\frac{d^{1/2}}{dx^{1/2}}$ represents the fractional derivative operator of order 1/2, $\Gamma$ denotes the gamma function, and $a$ is a constant Which was further elaborated for the following representation of $f(x)$ by Fourier

$$f(x) = \frac{1}{2\pi} \int_{-\infty}^{\infty} f(\eta)d\eta \int_{-\infty}^{\infty} \cos\theta(x-\eta)d\theta. \tag{2}$$

where η and θ are variables. Using the above representation, we can define the fractional derivative of order β for a function f(x) as follows:

$$\frac{d^{\beta}f(x)}{dx^{\beta}} = \frac{1}{2\pi}\int_{-\infty}^{\infty}f(\eta)d\eta\int_{-\infty}^{\infty}\theta^{\beta}cos(\theta(x-\eta)+\frac{\beta\pi}{2})d\theta.$$ (3)

where β is an arbitrary constant.

This definition of fractional calculus is used to solve the combined equation arising from the tautochrone problem, where we seek to determine an unknown function f(γ) by evaluating the following integral:

$$k = \int_{0}^{x}(x-\gamma)^{\alpha}f(\gamma)d\gamma$$ (4)

where α is a constant and γ is a variable.

By setting f(γ) equal to $\sqrt{\pi}\left[\frac{d^{-1/2}f(x)}{dx^{-1/2}}\right]$ and using this expression on both sides of Eq (4), we can obtain:

$$\frac{d^{1/2}k}{dx^{1/2}} = \sqrt{\pi}f(x)$$ (5)

where $\frac{d^{1/2}}{dx^{1/2}}$ denotes the fractional derivative operator of order 1/2.

Under appropriate conditions for $f(x)$, fractional operators have the property that:

$$\frac{d^{1/2}}{dx^{1/2}}\left[\frac{d^{-1/2}f(x)}{dx^{-1/2}}\right] = \frac{d^{0}f(x)}{dx^{0}} = f(x).$$ (6)

The first formal and clear definition of partial acquisition was given by Liouville [10–12] in 1832. Liouville extended the function f(x) as a series and assumed that for order numbers $\beta$ where the series changes, the following expression holds:

$$f(x) = \sum_{0}^{\infty}p_{n}e^{q_{n}x}$$ (7)

and assumed

$$D^{\beta}f(x) = \sum_{0}^{\infty}p_{n}q_{n}^{\beta}e^{q_{n}x}$$ (8)

where $D^{\beta}$ denotes the fractional derivative operator of order $\beta$ and $e$ is the exponential function.

Later on, a symbolic method for solving different line numbers and constant coefficients using fractional calculus, Riemann has proposed the following definition of fractional integration

$$D^{-\beta}f(x) = \frac{1}{\Gamma(\beta)}\int_{p}^{x}(x-\gamma)^{\beta-1}f(\gamma)d\gamma + \varphi(x)$$ (9)

where $\varphi(x)$ is Riemann's complementary function.

Although Fractional Calculus has been in the literature for more than 300 years, its uses have only recently been discovered. Many scientists are currently working with fractional calculus and their use in various fields.

## 2.1. Fractional order derivative definitions

The fractional order derivatives are generalization of the integer-order derivatives, where the order of the derivative is a non-integer value. There are several definitions of fractional derivatives, among them are the Grunwald-Letnikov and the Riemann-Liouville definitions.

### 2.1.1. Grunwald-Letnikov fractional derivative

The Grunwald-Letnikov fractional derivative is defined as the limit of a finite difference quotient, where the denominator is raised to the power of the non-integer order of the derivative. Successive differentiations of function $f(t)$ are given by

$$f^{(1)}(t) = \lim_{h \to 0} \frac{f(t+h) - f(t)}{h} \tag{10}$$

$$f^{(2)}(t) = \lim_{h \to 0} \frac{f^{(1)}(t+h) - f^{(1)}(t)}{h} = \lim_{h \to 0} \frac{f(t+2h) - 2f(t+h) + f(t)}{h^2} \tag{11}$$

etc. In general

$$f^{(n)}(t) = D^n f(t) = \lim_{h \to 0} \frac{1}{h^n} \sum_{k=0}^{n} (-1)^k \binom{n}{k} f(t - kh), \tag{12}$$

Where,

$$\binom{n}{k} = \frac{n!}{k!(n-k)!} \tag{13}$$

is a binomial coefficient. For a non-integer $\alpha > 0$, we can write

$$\binom{\alpha}{k} = \frac{\Gamma(\alpha + 1)}{k! \Gamma(\alpha - k + 1)}. \tag{14}$$

The Grunwald-Letnikov definition is the generalization of the definition to a non-integer $\alpha > 0$

$$D_a^\alpha f(t) = \lim_{h \to 0} \frac{1}{h^\alpha} \sum_{k=0}^{\frac{t-a}{h}} (-1)^k \frac{\Gamma(\alpha + 1)}{k! \Gamma(\alpha - k + 1)} f(t - kh), \tag{15}$$

Fractional integral of order $\alpha > 0$ is defined by

$$D_a^{-\alpha} f(t) = \lim_{h \to 0} \frac{1}{h^\alpha} \sum_{k=0}^{\frac{t-a}{h}} \frac{\Gamma(\alpha + 1)}{k! \Gamma(\alpha)} f(t - kh), \tag{16}$$

### 2.1.2. Riemann-Liouville fractional integral/derivative

Riemann-Liouville fractional integral operator is a direct generalization of the Cauchy's formula for an $n$-fold integral.

$$\int_a^x dt \int_a^{x_1} dt \ldots \int_a^{x_{n-1}} f(t) dt = \frac{1}{(n-1)!} \int_a^x \frac{f(t)}{(x-t)^{1-n}} dt \tag{17}$$

*If* f(x)$\epsilon C[a,b]$ and $a > 0$ then

$$I_{a^+}^\alpha f(x) = \frac{1}{\Gamma(\alpha)} \int_a^x \frac{f(t)}{(x-t)^{1-\alpha}} dt, \qquad x > a,$$

$$I_{b^-}^{\alpha}f(x) = \frac{1}{\Gamma(\alpha)} \int_x^b \frac{f(t)}{(x-t)^{1-\alpha}} dt, \qquad x < b,$$

are called as the left sided and the right sided Riemann-Liouville fractional integral of order $\alpha$, respectively.

### 2.1.3. Definition

Let $\alpha \epsilon$ (0,1), the

$$\mathbb{D}_a^{\alpha}f(x) = \frac{1}{\Gamma(1-\alpha)} \frac{d}{dx} \int_a^x \frac{f(t)}{(x-t)^{\alpha}} dt = DI_a^{1-\alpha}f(x) \tag{18}$$

is called the left side Riemann-Liouville fractional derivative of order $\alpha$ when RHS exists.

### 2.1.4. Definition

Let $n - 1 < \alpha \leq n$ then the left sided and right sided Riemann-Liouville fractional derivatives of order $\alpha$ are defined as:

$$\mathbb{D}_{a^+}^{\alpha}f(x) = \frac{1}{\Gamma(1-\alpha)} \frac{d^n}{dx^n} \int_a^x \frac{f(t)}{(x-t)^{\alpha+1-n}} dt = D^n I_{a^+}^{n-\alpha}f(x), \; x > a \tag{19}$$

$$\mathbb{D}_{b^-}^{\alpha}f(x) = \frac{1}{\Gamma(1-\alpha)} \frac{d^n}{dx^n} \int_x^b \frac{f(t)}{(x-t)^{\alpha+1-n}} dt = D^n I_{b^-}^{n-\alpha}f(x), \; x < b \tag{20}$$

The Riemann-Liouville fractional derivative of constant is not zero.

$$\mathbb{D}^{\alpha}C = \frac{Ct^{-\alpha}}{\Gamma(1-\alpha)} \neq 0 \tag{21}$$

Initial value problem (IVP) containing Riemann-Liouville fractional derivative requires initial conditions of the form $\mathbb{D}^{\alpha-j}f(0)$ i.e

$$I^{\alpha}(\mathbb{D}^{\alpha}f(t)) = f(t) - \sum_{j=1}^n \mathbb{D}^{\alpha-j}f(0) \frac{t^{\alpha-j}}{\Gamma(\alpha-j+1)}, \quad n-1 \leq \alpha < n \tag{22}$$

which are not useful in real phenomena. To overcome these drawbacks, M. Caputo proposed a new definition of derivatives which allows the initial conditions of fractional IVPs to be formulated in a form that involves only the boundary values of full-order derivatives in the lower terminal.

### 2.1.5. Caputo fractional derivative

Caputo fractional derivative is defined as:

Let $f \epsilon C^n[a, b]$ *and* n-1$< \alpha <$n*then*

$$D_a^{\alpha}f(x) = DI^{n-\alpha}D^n f(x) = \frac{1}{\Gamma(\alpha-n)} \int_a^x \frac{f^{(n)}(t)}{(x-t)^{(\alpha-n+1)}} dt, \; a < x < b \tag{23}$$

Properties

$$D_a^\alpha C = 0, \ C \text{ is constant}$$

$$\lim_{\alpha \to n} {}^c D_a^\alpha f(x) = f^n(x)$$

### 2.1.6. Relation between Riemann-Liouville and Caputo derivatives

Theorem

Let $f \epsilon C^n[a, b]$ *and n-1$<\alpha<$n then*. Then R-L and Caputo fractional derivatives are connected by the relation:

$$\mathbb{D}_a^\alpha f(x) = D_a^\alpha f(x) + \sum\nolimits_{k=0}^{n-1} \frac{f^{(k)}(a^+)}{\Gamma(1 + k - \alpha)} (x - a)^{k-\alpha} \tag{24}$$

Proof

$$\mathbb{D}^\alpha f(x) = D^n I^{n-\alpha} f(x) = D^n \left[ I^{n-\alpha} \left( I^{(n)} f^{(n)}(x) + \sum\nolimits_{k=0}^{n-1} \frac{f^{(k)}(a^+)}{k!} (x - a)^k \right) \right]$$

$$= I^{n-\alpha} f^{(n)}(x) + D^n I^{n-\alpha} \sum\nolimits_{k=0}^{n-1} \frac{f^{(k)}(a^+)}{k!} (x - a)^k$$

$$= D_a^\alpha f(x) + \sum\nolimits_{k=0}^{n-1} \frac{f^{(k)}(a^+)}{\Gamma(1 + k - \alpha)} (x - a)^{k-\alpha}$$

From above following results can obtained

If $\alpha = n \epsilon N, \ then \mathbb{D}_a^\alpha f(x) = D_a^\alpha f(x) = D^n f(x)$

If $f^{(a)}(a) = 0 \ for \ k = 0, 1, \ldots. n - 1, then \mathbb{D}_a^\alpha f(x) = D_a^\alpha f(x)$

If $0 < \alpha < 1$, then $\mathbb{D}_a^\alpha f(x) = D_a^\alpha f(x) + \frac{f(a)}{\Gamma(1-\alpha)} (x - a)^{-\alpha}$

### 2.1.7. Theorem

Let $f \epsilon C^n[a, b]$ *and* n-1$<\alpha<$n then

$$I_a^\alpha D_a^\alpha f(x) = f(x) - \sum\nolimits_{k=0}^{n-1} \frac{f^{(k)}(a^+)}{k!} (x - a),^k x \geq a \tag{25}$$

Proof

$$I_a^\alpha D_a^\alpha f(x) = I_a^\alpha I_a^{n-\alpha} f^{(n)}(x) = I^{(n)} f^{(n)}(x) = f(x) - \sum_{k=0}^{n-1} \frac{f^{(k)}(a)}{k!} (x - a)^k, x \geq a$$

In particular case of the more general property

$$I_a^\alpha D_a^\beta f(x) = I_a^\alpha I_a^{m-\beta} f^m(x) = I_a^{\alpha-\beta} (I^{(n)} f^{(n)}(x)) \alpha > \beta$$

$$= I_a^{\alpha-\beta} f(x) - \sum\nolimits_{k=0}^{n-1} \frac{f^{(k)}(a)}{\Gamma(\alpha-\beta+k+1)} (x - a)^{\alpha-\beta+k}, x \geq a, m - 1 < \beta < m.$$

## 3. Materials and methods

Laboratory grade UHMWPE resin powder with physical properties shown in Table 1 was purchased from Sigma-Aldrich. The powder was pressed into sheets using a Gibitre apparatus laboratory press at constant pressures of 200bar, while holding for 12–15 minutes at 150˚C,

**Table 1. UHMWPE properties.**

| Ultrahigh Molecular Weight Polyethylene (UHMWPE) | |
| --- | --- |
| Physical appearance | Polymer-resin |
| Formula (Empirical) | $(-CH_2-CH_2-)_v$ |
| Density ($\rho$) | 0.927 g/mL |
| Temperature (melting) | 130–136˚C |
| Molecular weight | 3–6 million gmol−1 |
| Melt flow rate (MFI) | 0.05 g/10 min |
| Physical state | Powder |

160˚C and 190˚C, respectively. The pressed samples were then cooled down to room temperature (25˚C) under the same pressure as mentioned above. After cooling, the surfaces of all samples were cleaned with acetone to remove any dirt or impurities. The thickness of each sample was measured with the help of IR vibration band at 2020cm−1 (an internal PE standard, which remains unaffected due to modification within the structure in PE matrix). The thickness of each sample was approximately 1mm.

Subsequent to preparations, the sheets were treated with 50 and 25-kGy of gamma-dose at 25˚C in open air with Co-60 gamma source from Pakistan radiation services Lahore at a constant dose rate of 1.02 kGy/h. For identification, the samples were labelled with codes P followed by absorbed dose values i.e. P-50, P-25 and P-0 respectively. According to reported literature, the radiation modification in UHMWPE is usually reflected in the following three regions of IR spectra

- 800–1100 $cm^{-1}$ which belongs to vinyl and trans-vinyldene absorption

- 1680–1800 $cm^{-1}$ which belongs to–C = O containing products

- 3300–3600 $cm^{-1}$ which belongs to hydrogen bonded and un bonded peroxide products

The experimental measurement were taken with the help of FTIR spectrometer (Nicolet-6700: Thermo Electron Corporation, Waltham, MA, USA) on the following setting

- Measurement Mode = Total attenuated reflection

- Measurement range = 4000 cm$^{-1}$ to 400 cm$^{-1}$

- Measurement resolution = 2 cm$^{1}$

- Measurement number of scans 216

- Measuring averages = 3 to 4 for each sample

- Measurement points = 3 to 4 for each

- Output representation mode = % reflectance with constant baseline correction

Subsequent to experimental measurement, the % transmittance data is plotted as a function of wave number and aforementioned regions of interest are zoomed to have idea about radiation induced modifications within the structure of UHMWPE.

After examining/analysing the experimental spectra, differential filters of various fractional orders (ranging from 0 to 1) are applied for removing the any possible background ground and quasi static noise. The data consists of experimental spectra which are considered as $0^{th}$ order (for each sample) followed by simulated spectra by applying $1^{st}$, $0.9^{th}$, $0.8^{th}$, $0.7^{th}$ and $0.5^{th}$ order differential transformation. To apply the differential transformation, the series

function that best fits the experimental data was obtained with the standard non-linear qua-
dratic curve fitting algorithm, namely the Levenberg-Marquardt (LM) algorithm. The LM
algorithm was used for the purpose in the course of this study as there is no effective numerical
model to solve the integral and derivative of fractional order, and the main topic of the
research was to apply the minimum transformation to the real experimental data to avoid
information loss as much as possible. i.e., lower fractional order differential transformation
was required. The fitting function obtained from series approximation fitting to experimental
data is used for simulating the FD ranging from 0 to 1.

For the calculation of FD, a MATLAB-function is written that used parameters obtained
from curve fitting for the generation of y-values for a given x-ones. After generating the values,
it calculates the fractional-order derivatives using the classical-definition as proposed by Rie-
mann-Liouville (RL). Lagrange-operator based law is used in this definition, where the $\alpha^{th}$
order rate of change resulted by taking $t^{th}$ derivative over integral of $(t-\alpha)^{th}$ order. It satisfies
the integral condition $n > \alpha$. Beginning of the definition of the integral RL (fractions) $0 < \alpha < 1$
and the initial value a = 0.

$$\mathcal{J}_{a^+}^{\alpha} f(t) = \frac{1}{\Gamma(\alpha)} \int_a^t (t - \tau)^{\alpha-1} f(\tau) d\tau \tag{26}$$

The values were calculated for $\alpha$ = 0.5, 0.7, 0.8, 0.9, and 1.

After simulating the FD, the correlation coefficients (CCs) were calculated while using the
relation given below. This was done to figure out the best transformation with minimum loss
of information. [34].

$$CC(XY) = \frac{\sum_{i=1}^n (X_i - \bar{X})(Y_i - \bar{Y})}{\sqrt{\sum_{i=1}^n (X_i - \bar{X})^2 \sum_{i=1}^n (Y_i - \bar{Y})^2}} \tag{27}$$

Here

- $\bar{X} \& \bar{Y}$ are the mean-values of both parameters

- The range of CC(XY) is from+1 to -1

In addition to aforementioned analysis-based point to point correlation of simulated data,
PCA was performed to see the dispersion behaviour of data on applying differential filters as a
function of order of filter. PCA is basically dimension reduction technique that transform the
original elements into new, uncorrelated factors (axes) known as principal components. The
new axis follows the direction of maximum variance. It is a linear combination of factors is
used to unveil the hidden patterns, dynamics, and correlation in the given set of data as well as
to standardize variables for comparison the dispersion in the set of data. In this type of analy-
sis, the original variables are divided into groups using a new set of components called axis
rotation in PCA. It's used to identify a sample's compositional influencing factor. PCA gives a
specific explanation for the most important element, which reflects the interpretation of the
entire data set. With the least loss of information, PCA summarizes the statistical association
between the order of the differential transformation. Principal component analysis is
expressed using the following equation:

$$Zij = a_{i1}x_{1j} + a_{i2}x_{2j} + \cdots + a_{im}x_{mj} \tag{28}$$

where $Z$ is the component score, $a$ is the component loading, $x$ is the measured value of a vari-
able, $i$ is the component number, $j$ is the sample number, and $m$ is the total number of

variables. The score values (for each sample in all region of interest) belongs to first two PCs were plotted for the analysis of simulated data FD data.

After figuring out the suitable order of transformation from the CCs and PCA analysis, the classification of pristine and irradiated samples is performed while using Cluster Analysis. Cluster analysis is a multivariate technique whose main function is to provide tools for constructing objects based on unique parametric features. In cluster analysis, the basic approach for categorizing items is that one object is comparable to the other in terms of established descriptive criteria. To distinguish between important and irrelevant variables, the cluster object's results correlate to strong internal homogeneity and extremely high exterior heterogeneity. As a result, conceptual considerations must be used to justify variables in cluster analysis. Hierarchical agglomerative clustering or Hierarchy Cluster Analysis (HCA) is a common method for evaluating similarities between large set of data, and it is typically represented by a dendrogram. The major goal of this study was to determine the similarity and dissimilarity between each sample of UHMWPE for all three regions of interest while calculating connection between individual clusters. It is therefore, the parametric mean value for each spectral region of interest was represented by a dendrogram and further analyzed for identification of radiation induced modifications. The research methodology of this study is summarized in the flow diagram shown in Fig 1

## 4. Results and discussion

### 4.1.Experimental results

The experimental data is given in Fig 1 where Fig 2(A) is representing the full FTIR spectra from 400–4000 cm$^{-1}$ of all three samples and the Fig 2(B) is representing the three regions of interest that are significantly influenced with irradiation treatment. From the above figure it is clearly evident that FTIR spectrum shows increase in absorption in the following bands after radiation.

- The modification in the IR region ranging from $800 cm^{-1}$ to $1100\ cm^{-1}$ is evident on for irradiated samples, and this modification is attributed to conversion of C = C un-saturation to vinylidene ($R_1R_2C = CH_2$) and trans-vinylene (–CH = CH–) [13–16].

- The radiation induced oxidation reactions results in increase in absorption of carbonyl C = O species, and this is reflected from the % decrease of transmittance from 1650 $cm^{-1}$ to 1850 $cm^{-1}$ for irradiated samples i.e. P-25 and P-50 (see Fig 2(B))[15–18]

- Increase in absorption from 3000 $cm^{-1}$ to 3600 $cm^{-1}$ corresponding to typical oxidation of polyethylene (PE) end products i.e., hydrogen bonded and unbounded hydro peroxide[16, 19, 20].

UHMWPE is a polyethylene polymer that can contain small amounts of other functional groups, such as vinyl or acetyl groups, which can introduce distinctive vibrational modes in the IR spectrum. Vinyl groups usually exhibit characteristic IR absorption bands, such as C-H bend at around 960–970 cm$^{-1}$ [16, 21–24]. Gamma irradiation of UHMWPE can result in radiation-induced degradation, which involves the breaking of carbon-carbon double bonds and the formation of new chemical species. It has been reported that trans-vinylene groups are formed as a result of gamma irradiation, and these can be detected by FTIR spectroscopy as a weak peak at around 965 cm$^{-1}$ [13, 14, 17, 22, 24, 25]. In addition, it is possible for vinyl groups to convert to vinylidene groups during irradiation while giving the absorption signal at around 913 cm$^{-1}$ which belongs to the C-H bending vibration of vinylidene [24, 26, 27]. The increase

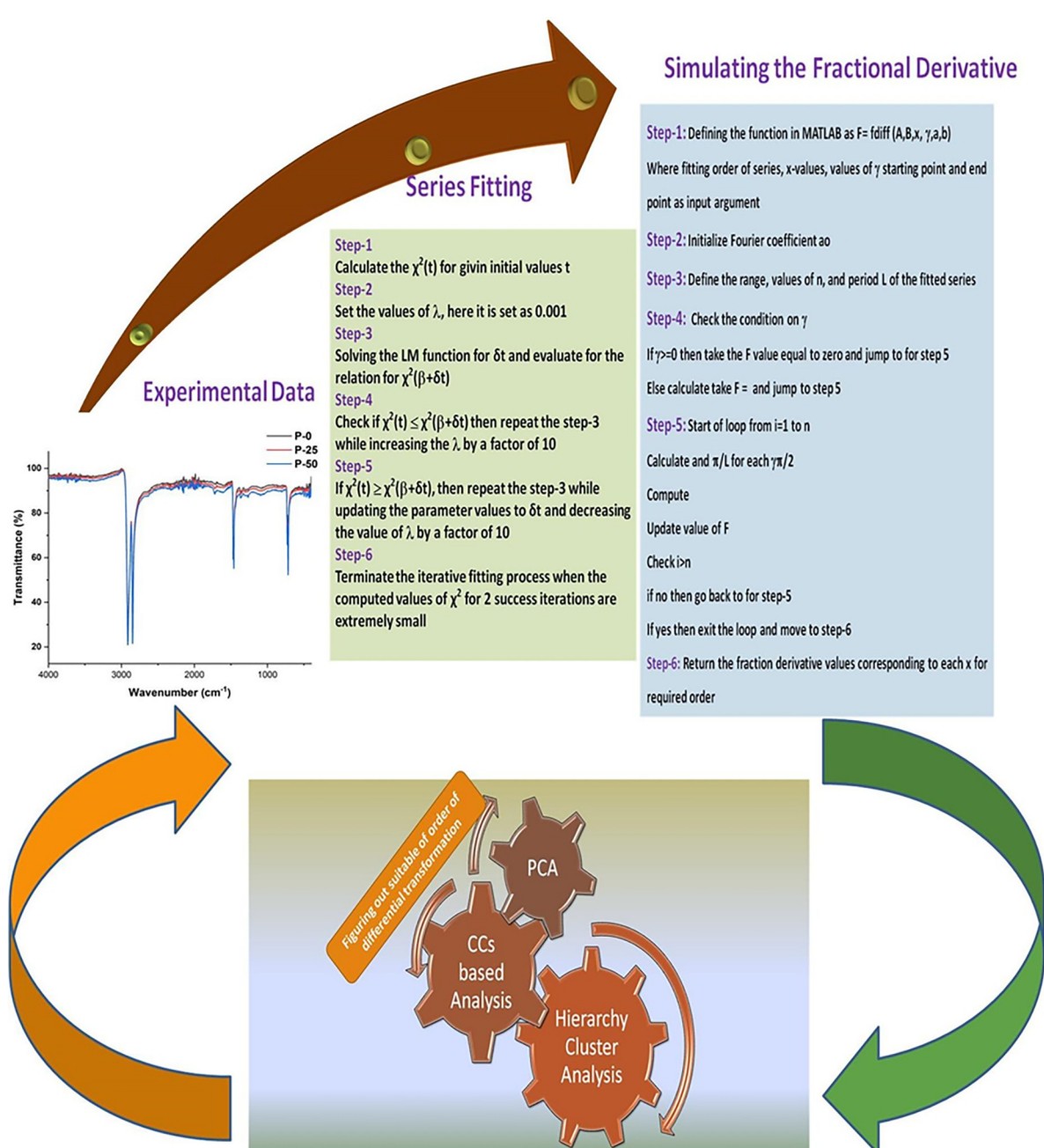

**Fig 1. Block diagram of spectral differentiating ad pre-processing adopted during this study.**

in absorption from 1700–1740 belongs to oxidation products with carbonyl (-C = O) functional groups [27] formed during the oxidation chain reactions. Furtherance to it, radiation is responsible for producing the vinylidene ($R_1R_2C = CH_2$) and trans-vinylene (–CH = CH–) because of radiation induced free radicals reactions on main polymer chain [22, 25]. The absorbance due to vinylidene ($R_1R_2C = CH_2$) and trans-vinylene (–CH = CH–) usually appears at 920–930 $cm^{-1}$ and 960–980 $cm^{-1}$ but usually it is not visible in AT-FTIR experimental spectra because of weak signals [25, 28–32]. It is therefore, experimental spectra shown in Fig 2(B) has no clear indication of absorbance signals for vinylidene ($R_1R_2C = CH_2$) and trans-

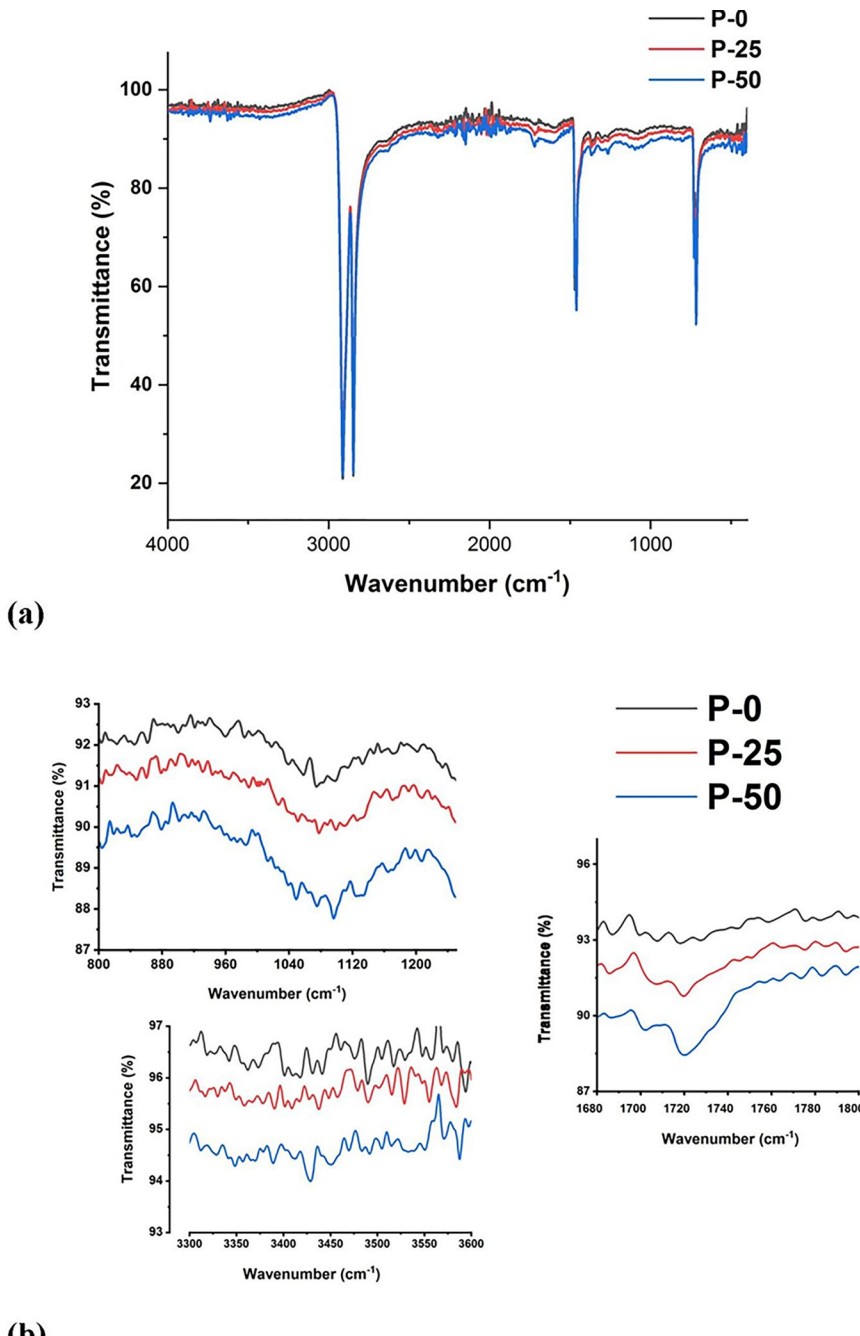

**Fig 2.** (a) & (b). FTIR spectra different sample of UHMPWPE.

vinylene (–CH = CH–). In order to quantify the concentration abovementioned weak bands, experimental measurement of thin slices of UHMWPE in absorption mode has been performed. Subsequent to measurements, normalizations with respect to internal standards is required as well. It is also worth to mentioned here that the according to the literature, usually the concentrations of weak bands i.e. vinylidene, methyl groups and trans- vinylene are almost negligible for pristine UHMWPE [23, 31].

## 4.2. Simulated fractional derivatives

Succeeding the step wise procedure discussed in the section 3 and represented in Fig 1, MATLAB is used to pre-process the experimental spectral data of all samples under investigation, and to obtained the transformed data after apply fractional order differential transformation. The series function that is fitted to experimental data is passed through differential filtered of various orders ranging from 0 to 1. The data for PE-0, P-25 and PE-50 for all three regions of interest i.e., 940–960 cm$^{-1}$, 1700–1750 cm$^{-1}$, and 3300–3600 cm$^{-1}$ are represented as 3d water fall plots shown in Fig 3(3A–3C). The Y-axis of the 3d waterfall diagrams corresponds to the differential values after applying the transformations, and wavenumber in cm$^{-1}$ is representing as X-values. The As a result of the variation of the reflectance values, as a characteristic of the number of waves and the irradiation treatment, the following main changes are worth to be highlighted.

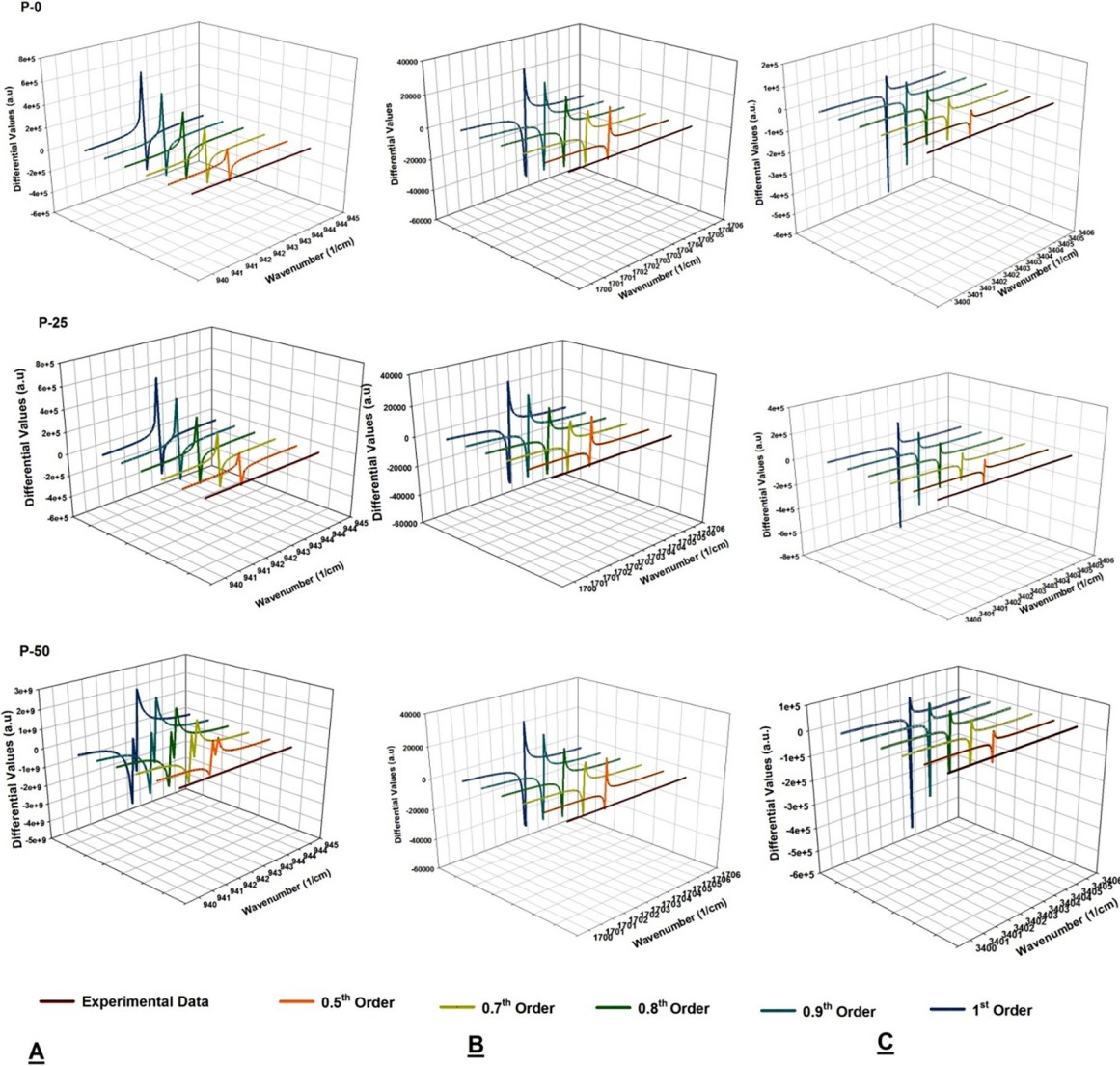

**Fig 3.** Simulated Fractional order and 1$^{st}$ order derivative plots from 940–960 cm$^{-1}$ (A) 1700–1750 cm$^{-1}$ (B) and 3300–3600 cm$^{-1}$ (C) for P-0, 25, and 50 kGy irradiated samples.

- A significant increase in differential maxima and minima is evident with the order of transformation and the value of absorbed dose for all spectra investigated in the study.

- Almost all the spectra have singlet for 1700–1750 cm$^{-1}$ and 3300–3600 cm$^{-1}$, however; in the region 940–960 cm$^{-1}$, the differential transformation of experimental spectrum of sample labeled with P-50 reveals the doublet for all orders (please see the 3d water fall plot of P-50 in Fig 3(A)).

- For in depth investigation of spectrum which is showing doublet on transformation, integration of differential spectra from 800–1100 cm$^{-1}$ for P-50 sample is taken which reveals the two peaks at 913 cm$^{-1}$ and 975 cm$^{-1}$ for all orders, respectively (see Fig 4)

The increase in the differential values with respect to dose for all regions reveals the fact that each sample suffers on receiving the radiation treatment. This results in increase of oxidation products having–C = O functional groups and hydro-peroxides products. Moreover, the modifications are evident and same for all samples even at lowest order of transformation used in the study i.e. 0.5$^{\text{th}}$ order as well. The splitting of peak in the region 940–960 cm$^{-1}$ for 50 kGy irradiated sample is basically due conversion vinyl in vinylidene ($R_1R_2C = CH_2$) and trans-vinylene (–CH = CH–) because of gamma irradiation. The peak at 913 cm$^{-1}$ and 975 cm$^{-1}$ which are observed in FTIR spectra of 50 kGy irradiated sample belongs to vinylidene ($R_1R_2C = CH_2$) and trans-vinylene (–CH = CH–), respectively. This is confirmation of conversion vinyl groups because of their interaction of with irradiation that is responsible for various

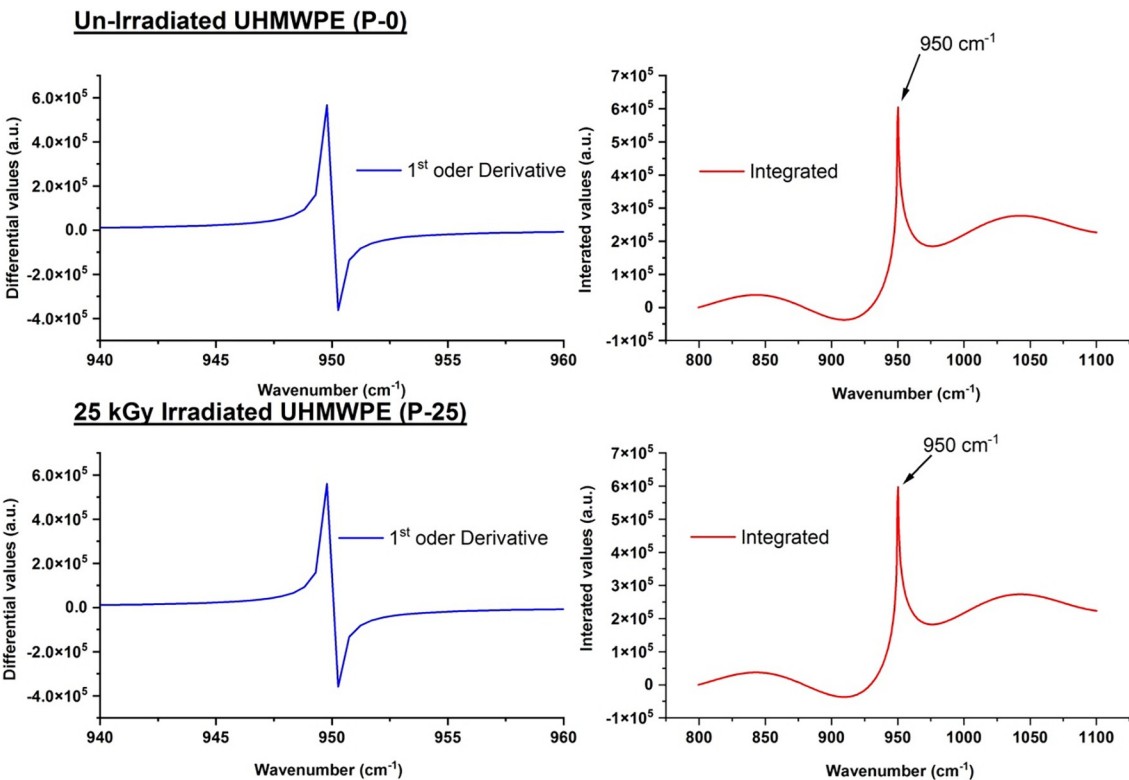

**Fig 4.** (I) ATIR-FTIR spectra of sample labelled as P-0 (A), after applying 1st order differential filter (B), and integrated of filtered curve for indemnification of peak positions (II) ATIR-FTIR spectra of sample labelled as P-25 (A), after applying 1st order differential filter (B), and integrated of filtered curve for indemnification of peak positions.

reactions, including the formation of different species such as vinylidene and trans-vinylene. The mechanism of these reactions is complex and depends on various factors, including the type and energy of the radiation, the structure and composition of UHMWPE, and the reaction conditions. Irradiation-induced reactions in polymers involve the transfer of energy from the radiation to the polymer chains, leading to the formation of free radicals. These free radicals can react with vinyl groups to form vinyl radicals, which can then undergo rearrangement to form vinylidene and trans-vinylene groups. The reaction schemes are as follows:

UHMWPE + high energy radiation (gamma, electron beam, etc.) → formation of free radicals ($^{\bullet}CH_2$, $^{\bullet}C(CH_3)_2$, etc.)

$$^{\bullet}CH_2 + CH_2 = CH_2 \rightarrow CH_2 = CH - CH_2^{\bullet}$$

$$CH_2 = CH - CH_2 \bullet \rightarrow CH_2 = C = CH_2 + H^{\bullet}$$

$$^{\bullet}C(CH_3)_2 + CH_2 = CH_2 \rightarrow C(CH_3) = CH - CH_2^{\bullet}$$

$$C(CH_3) = CH - CH_2^{\bullet} \rightarrow CH_2 = C(CH_3) - CH = CH_2 + H^{\bullet}$$

Vinylidene and trans-vinylene groups can be detected by FTIR spectroscopy through their characteristic absorption peaks. The C = C stretch for trans-vinylene is observed around 965 $cm^{-1}$, while the C = C stretch for vinylidene is observed around 1600–1640 cm-1, with the C-H bend observed around 1000–950 cm-1. In un-irradiated UHMWPE, the C-H bend of vinyl is typically observed at around 950 $cm^{-1}$ which is clearly evident on applying the differential order filters to FTIR spectra of un-irradiated UHMWPE (see Fig 4 (I)). The results found in this study are consisted with the peak positions of vinyl, vinylidene and trans-vinylene with the literature [23, 25, 28–34], thus proves the potential of proposed methodology for subject matter of interest.

Form the results shown in Figs 3 and 4; it is clear that applying differential filters to ATR-FTIR experimental spectra is helpful in direct detection of trans-vinylene even at lower order used in this study i.e., 0.5th. However; still, it is debatable that which order of filter is more suitable for application to ATR-FTIR experimental spectra. It is therefore; sensitivity and specificity of each order of transformation for all regions of interest need to be investigated before moving for using the data to assess the modification induced by irradiating the UHMWPE with 25 and 50 kGy of gamma dose. For this purpose, analysis based on point-to-point linear correlations index model and PCA is executed. The results of the analysis are discussed in the next section.

### 4.3. Sensitivity and specificity analysis of fractional orders

**4.3.1. Correlation index based sensitivity analysis.**   Although, the strength of CCs is higher for experimental data (i.e. CCs data of 0th order spectrum) but the main interest of the study is to see the behaviour of CCs for spectra after applying the differential filters. For better quantitative examination of radiation induced spectra alternations within the UHMWPE matrix, it is necessary to identify the sensitive order of differential transformation of spectra data that are responsible for refining the accurateness of the model. For the analysis, point-to-point linear correlations strength between the wave numbers and differential values (belongs to each transformation) for all three regions are plotted in Fig 5(I) a-c. The plotted values of CCs are of P-0 sample and calculated while using the relation given in section 3. In the figure the strength of correlation coefficients (CCs) for each wavenumber belonging to each order is plotted as parallel plot for better visualization of point-to-point correlation of spectral data i.e.,

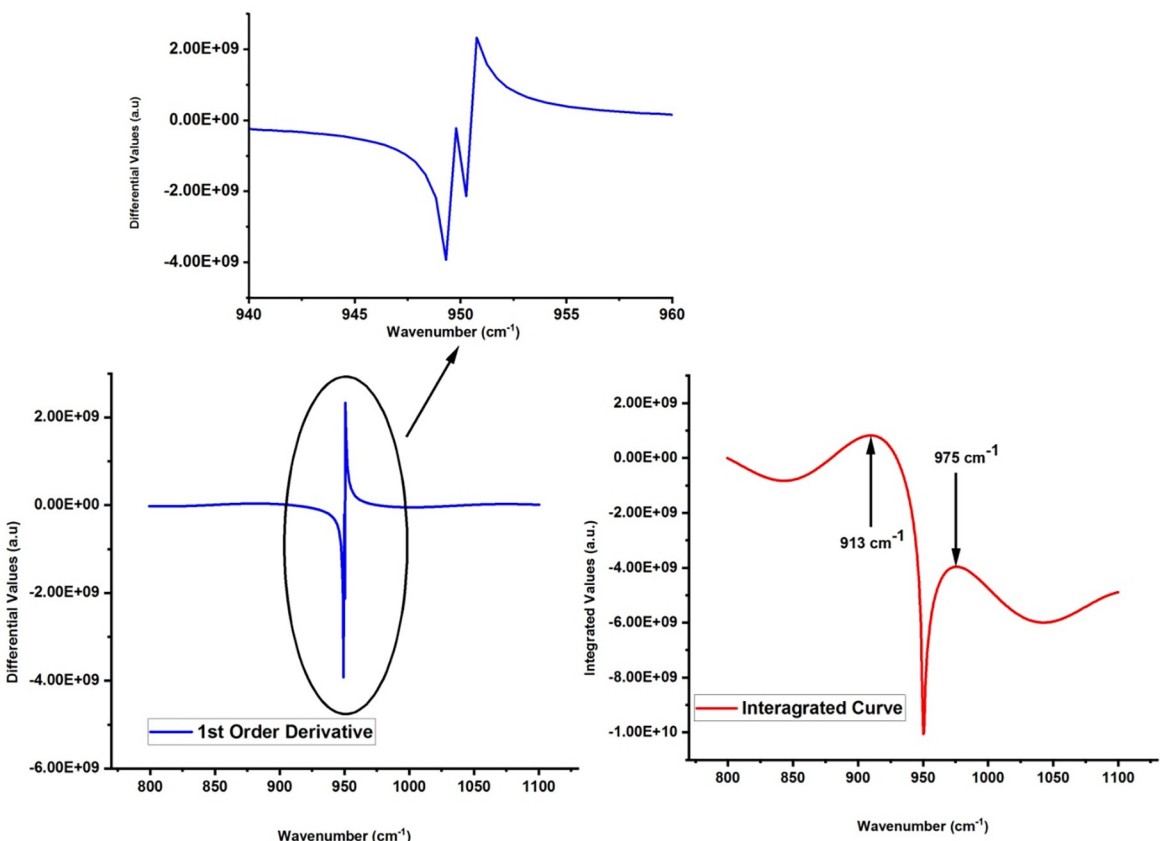

**Fig 5.** ATIR-FTIR spectra of sample labelled as P-50 Experimental (A), after applying 1st order differential filter (B), and integrated of filtered curve for indemnification of peak positions.

experimental and simulated fractional data. In the type of data representations shown in Fig 5, the more convergent and stronger distribution of CCs is the indication of negligible contribution of quasi static and background noise with minimum loss of information during transformation of experimental spectral data and vice versa. It can be seen from the Fig 5 that that the strength of CCs for differential transformation of $0.5^{th}$ order is higher, and the distribution is more convergent over the spectral range as compared to others differential orders. In addition, this factor is true for all three regions of interest i.e. 800–1100 cm$^{-1}$, 1680–1800 cm$^{-1}$ and 3300–3600 cm$^{-1}$. Furthermore, the strength and convergence of CCs has decreasing trend on moving toward the higher order of differentiation for all spectral regions of interest (see Fig 5 (I)). These findings assure the fact that applying differential filters of lower order ($0.5^{th}$ in the course of current study) is more appropriate because of strength and point to point distribution of CCs over the spectral ranges of interest.

**4.3.2. Principal component analysis (PCA).** For further confirmation about the sensitivity of the order of differentiation, Principal Component Analysis (PCA) of spectral data (experimental and simulated FD) has been performed. The dispersion of data for un-irradiated UHMWPE sample is plotted as score plots on $1^{st}$ two principal components for all spectral regions of interest (see Fig 5 (II)). Following are the important insights revealed by the PCA analysis

- In all three regions, $1^{st}$ PCs are responsible for explaining more than 95% of variance in data

- The score value for data after applying 1st order differential filter fall in the opposite quadrant of experimental one (i.e. 0th order), and this is true for all regions of interest.

- The distribution score values for the differential filters belonging to interval $0<\alpha<1$, where α stands for the order of filter, is not random on PCA plane. It follows specific parabolic trajectory for all regions of interest

- The aforementioned parabolic trajectories start from experimental data i.e. 0th order, ends at the score value of spectral data after applying 1st order differential filter. The data belonging to fractional orders filtering distributed on the trajectories.

- The score value of spectra after applying lower order (e.g., 0.5th order in this study) filter is close to experimental one for all region of interest (see Fig 5 (II))

The aforementioned hidden trends revealed by PCA points to the fact that applying differential transformation to spectral results in increase in data dispersion and it is totally dependent on order of transformation filters. This means lower is the order of transformation lower the data dispersion thus minimizing the risk of information loss during the transformation. It is therefore, lower order of differential filter (0.5th order in the course of this study) is more suitable for accurate information structure from the given spectral data. The hidden patterns reveal by PCA regarding the dispersion of spectra after applying differential filters are in total agreement with results of CCs shown as column plot in Fig 5 (I).

From the results of point-to-point correlation analysis and PCA, it is obvious that applying 0.5th order differential filter is the further appropriate choice for assessing the radiation induced modifications in UHMWPE. It is therefore, simulated FD data of 0.5th order along with experimental for classifying the sample and assessing radiation damage with the help of cluster analysis. The details are given in the following section.

## 4.4. Assessing the radiation modification

The dendrogram shown in Fig 6 which is constituted while considering the similarities and dissimilarities in experimental spectral data and filtered data with differential filter of 0.5th order disclose the following information about the effect of radiations on UHMWPE structure.

- In the region 800–1100 cm-1, experimental data of all three samples and FD data of P-0 and P-25 form one cluster and P-50 Simulated FD spectral data form a standalone cluster (see Fig 7(A)). This trend points to the fact that in this region a small modification within the structure of UHMWPE occurs only for 50 kGy irradiated sample which is only become noticeable on applying the differential filter or taking the 0.5th order derivative of experimental curve in the above-mentioned spectral region. This modifications is attributed to extremely weak absorption of trans-vinylene (–CH = CH–) which were absent in un-irradiated and 25 kGy irradiated samples [23, 25, 28–34].

- In the regions 1680–1800 cm$^{-1}$ and 3300–3600 cm$^{-1}$, samples forms two cluster one based on experimental spectral data and second one based simulated 0.5th FD spectral data (see Fig 7B and 7C). Both dendogram show a clear indication of structural modification of because of radiation induced modifications for 50 kGy irradiated sample.

- Carbonyl-containing functional groups such as ketones, aldehydes, and carboxylic acids typically exhibit strong IR absorption bands in the region around 1700–1750 cm$^{-1}$ due to their C = O stretching vibrations. In contrast, hydroxyl (-OH) functional groups typically exhibit weaker IR absorption bands in the region around 3200–3600 cm$^{-1}$ due to their O-H

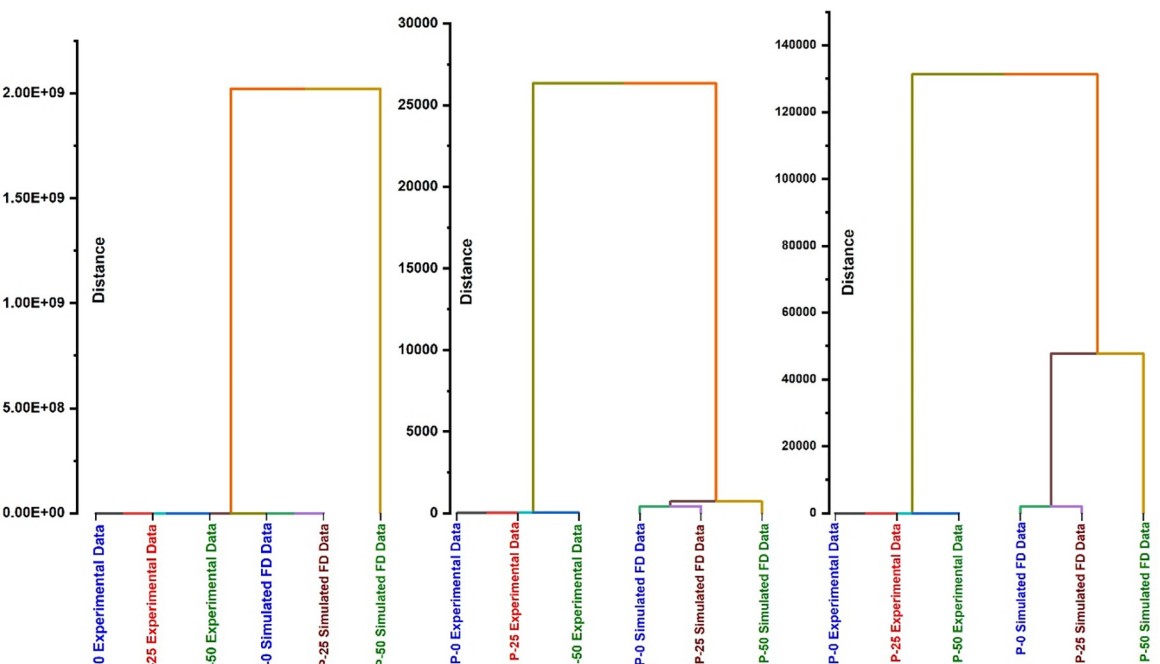

**Fig 6.** Dendrograms for grouping the samples on the basis of similarities in experimental spectra and filtered spectra with differential filter of 0.5th order for IR spectral region (A) 800–1100 cm$^{-1}$, (B) 1680–1800 cm$^{-1}$ and(C) 3300–3600 cm$^{-1}$.

stretching vibrations. Therefore, it is expected that the response of the carbonyl-containing functional groups would be higher than that of the hydroperoxide (-OOH) functional groups in terms of their IR absorption bands. This is because the carbonyl-containing functional groups typically have stronger IR absorption bands than the hydroperoxide functional groups.

## 5. Conclusion

In this paper fractional differential transformation from 0.5$^{th}$ to 1$^{st}$ order is successfully employed on the ATR-FTIR experimental spectra of pristine, 25 kGy and 50 kGy irradiated UHMWPE samples. The application of differential transformation is applied in three regions of IR spectra i.e. 800–1100 cm$^{-1}$, 1680–1800 cm$^{-1}$ and 3300–3600 cm$^{-1}$ respectively. The transforming the experimental data enables successfully unveils and distinguishes the absorption of weak bands of vinylidene (R$_1$R$_2$C = CH$_2$) and trans-vinylene (–CH = CH–), respectively for 50 kGy irradiated sample, thus reflecting efficacy of applying the differential transformation to experimental data of any order in the above mentioned range. However, the analyzing the strength and dispersions of point to point CCs for each order of transformation concludes the fact the lower is the order, higher is the information precision as reveals by spectral data, lower is dispersion if simulated FD spectra data, thus minimizing the risk of information loss. This sensitivity of also confirmed by 2D PCA plots which clearly reveals that the differential transformation for all region follow a parabolic path staring from experimental in one quadrant of plot, ending on 1$^{st}$ order in opposite quadrant. The fractional orders are lying on the trajectory with lowest one is at right next neighbouring position of experimental data, and affirms the using of 0.5$^{th}$ order differential transformation for assessing the radiation damage. Finally, the experimental data and 0.5$^{th}$ order simulated FD data is used as input for HCA analysis which

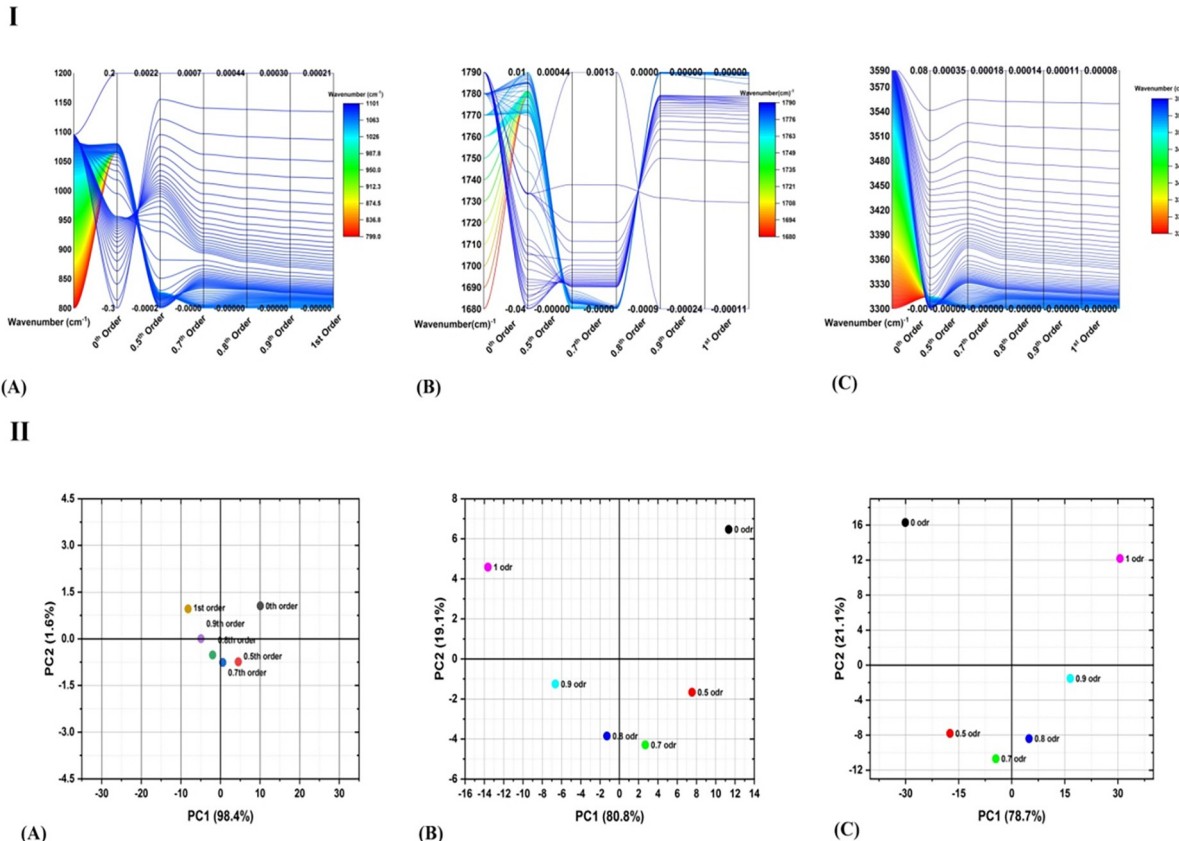

**Fig 7.** (I) Coefficients of point-to-point correlations of experimental and simulated FD spectral data for (A) 800–1100 cm$^{-1}$, (B) 1680–1800 cm$^{-1}$ and (C) 3300–3600 cm$^{-1}$. (II) Prncipal Component Anlsyis of experimental and simulated FD spectral data for (A) 800–1100 cm$^{-1}$, (B) 1680–1800 cm$^{-1}$ and (C) 3300–3600 cm$^{-1}$. Note: These results of Fig 5 are only for P-0 Sample i.e. for un-irradiated UHMWPE sample.

clearly concludes that applying differential filter is helpful for figuring out extremely small dissimilarities in spectra of sample because of radiation modifications.

## Author Contributions

**Conceptualization:** Malik Sajjad Mehmood, Muhammad Muddassar.

**Data curation:** Muhammad Mudassir Saeed, Malik Sajjad Mehmood.

**Formal analysis:** Muhammad Mudassir Saeed, Malik Sajjad Mehmood, Muhammad Muddassar.

**Investigation:** Muhammad Mudassir Saeed, Malik Sajjad Mehmood, Muhammad Muddassar.

**Methodology:** Muhammad Mudassir Saeed, Muhammad Muddassar.

**Resources:** Muhammad Mudassir Saeed, Malik Sajjad Mehmood.

**Software:** Muhammad Mudassir Saeed, Malik Sajjad Mehmood.

**Supervision:** Muhammad Muddassar.

**Validation:** Muhammad Muddassar.

**Writing – original draft:** Muhammad Mudassir Saeed, Malik Sajjad Mehmood.

**Writing – review & editing:** Malik Sajjad Mehmood, Muhammad Muddassar.

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
