## [Decision Letter · Decision Letter 0]

27 Feb 2023

PONE-D-22-32699Fractional order ATR-FTIR Differential Spectroscopy for Detection of Weak Bands and Assessing the Radiation Modifications in Gamma Sterilized UHMWPEPLOS ONE

Dear Dr. Muddassar,

Thank you for submitting your manuscript to PLOS ONE. After careful consideration, we feel that it has merit but does not fully meet PLOS ONE’s publication criteria as it currently stands. Therefore, we invite you to submit a revised version of the manuscript that addresses the points raised during the review process. Please submit your revised manuscript by Apr 13 2023 11:59PM. If you will need more time than this to complete your revisions, please reply to this message or contact the journal office at plosone@plos.org. Please include the following items when submitting your revised manuscript:A rebuttal letter that responds to each point raised by the academic editor and reviewer(s). You should upload this letter as a separate file labeled 'Response to Reviewers'.A marked-up copy of your manuscript that highlights changes made to the original version. You should upload this as a separate file labeled 'Revised Manuscript with Track Changes'.An unmarked version of your revised paper without tracked changes. You should upload this as a separate file labeled 'Manuscript'.

We look forward to receiving your revised manuscript.

Kind regards,

Hannes C Schniepp, Dr. sc. nat.

Academic Editor

PLOS ONE

Journal Requirements:

Reviewers' comments:

Reviewer's Responses to Questions

**Comments to the Author**

1. Is the manuscript technically sound, and do the data support the conclusions?

Reviewer #1: No

Reviewer #2: No

Reviewer #3: Yes

2. Has the statistical analysis been performed appropriately and rigorously? 

Reviewer #1: No

Reviewer #2: N/A

Reviewer #3: Yes

3. Have the authors made all data underlying the findings in their manuscript fully available?

Reviewer #1: Yes

Reviewer #2: Yes

Reviewer #3: Yes

4. Is the manuscript presented in an intelligible fashion and written in standard English?

Reviewer #1: No

Reviewer #2: No

Reviewer #3: Yes

5. Review Comments to the Author

Reviewer #1: The paper regards a particular application of non integer order system. The literature today includes more application about. Thspecific application regards the an application in spectroscopy. The paper is appropriate for a conference in the specific topics. the results agree but are considered only as a particular exercise. New analytical theories are not included and therefore classical strategies have been presented. The application is possibli only for specialists absolutely is not addressed for the PLOS ONEjournal. Indeed the discussion is unclear and the paper is not appealing for a wide class of readers.

Reviewer #2: The manuscript presents a method for the manipulation of ATR-FTIR spectra which, in the intention of the authors, should improve the sensitivity in the quantification of weak intensity bands; the method was applied to UHMWPE samples to investigate the modifications induced by irradiation with high energy radiations.

In my opinion, the study suffers from a fundamental flaw, which calls into question its entire validity. That is, however powerful the manipulation algorithm is, a reliable result cannot ignore the minimum sensitivity of the experimental measurements. For example, the authors aimed at quantifying, among others, the absorptions relating to trans-vinylene groups (965 cm-1), which are formed as a result of irradiation. The concentration of trans-vinylenes formed in UHMWPE, following gamma irradiation with doses of 25-50 kGy, has been verified to be lower than 10 mmol/kg [P. Bracco et al. Polymer. 46 (2005) 10648], while the molar extinction coefficient of this absorption is 168 l cm-1 mol-1 [De Kock R et al. J Polym Sci, Polym Lett 1964;2:339] and the penetration depth of the IR radiation at that wavelength varies approximately between 1 and 5 microns, depending on the refractive index of the crystal and the angle of incidence of the radiation, depending on the geometry of the system. Under these conditions, the Beer-Lambert law indicates that the absorbance of the trans-vinylene signal should be at most of the order of 8*10-4, corresponding to a 0.2% transmittance, which, in the spectra shown in Figure 2, is clearly indistinguishable from background noise. The same consideration applies to hydroperoxide groups, whose FTIR absorption falls around 3400 cm-1 in the form of a broad, weak peak.

An evidence of the inapplicability of the method to current spectra is, for instance, that the authors claim the appearance of signals relating to vinyl and trans-vinylene double bonds in the spectra of samples irradiated at 50 kGy, while it is well known from the literature that vinyl double bonds are present in the virgin material and disappear as a result of radical reactions triggered by irradiation and should therefore show just the opposite trend [J. Lacoste et al. Polym Degrad Stab. 34 (1991) 309]. Similarly, it is stated that “The modification in hydrogen bonded and unbounded hydro peroxide regions which belongs to typical end products of oxidations is higher as compared –C=O containing functional groups products”, while it should be exactly the opposite, given the lower response of the -OH stretching signal around 3400 cm-1, compared to the intense signal of the carbonyls around 1720 cm-1.

In view of these observations, I believe that the proposed method is not applicable in the present case. It would be different if the method were applied to spectra obtained in transmission mode, but in that case, the absorptions in question are generally well quantifiable even without manipulation.

In addition, the manuscript has several other weaknesses. The language is rather poor and would need a thorough revision, since sometimes the concepts themselves are not clear. The equations introduced in the long paragraph "Background literature" are not sufficiently commented, so as to allow the reader to follow the logical path, and many variables are undefined. The preparation of the samples is not adequately described: it is only reported that the UHMWPE powder was pressed to obtain 1 mm thick films; at what temperature, at what pressure and for what time? The bibliography is also poor: many statements would need adequate references which are not present, while about one third of the bibliography is made up of self-citations.

Reviewer #3: The article presents an important study on the use of fractional order differential transformation on ATR-FTIR spectra to identify and assess radiation modifications in gamma sterilized UHMWPE samples. The study aims to find a more sensitive and accurate approach to polymer characterization than existing methods such as FTIR, XRD, and DSC. The methodology explains the steps taken to obtain simulated spectra using differential transformation of fractional orders. The encouraging results obtained from the simulated spectra are analyzed using correlation index analysis, principal component analysis, and hierarchy cluster analysis. It is acceptable for publications after clarifying followings points and revising the manuscript in the light of these (minor revisions)

Introduction:

1. Why it is important to identify weak IR bands and radiation modifications in gamma sterilized UHMWPE? please add the importance of this identification

2. Revise the abstract while making the technical language in the abstract more accessible for non-experts?

3. Revise the introduction while providing the more detailed information on the challenges and limitations of current approaches to polymer characterization?

Materials and Methods:

1. What are the three regions of IR spectra where radiation modification in UHMWPE is usually reflected? Are there any other regions?? If so please mention

2. What type of algorithm was used for curve fitting in this study?

3. How were differential filters of various fractional orders applied in the study?

4. What is the purpose of calculating the correlation coefficients?

Results and Discussion

1. What is responsible for the increase in absorption of carbonyl C=O species?

2. Why are the peaks of vinylidene and trans-vinylene usually not visible in AT-FTIR experimental spectra?

3. What is the process of simulated fractional derivatives?

4. How is MATLAB used to pre-process experimental spectral data? Any special function or built-in module is used??

5. Why is it necessary to identify the sensitive order of differential transformation of spectra data?

6. What is the correlation index-based sensitivity analysis used for in this study? Please elaborate

7. What are CCs and how are they calculated in this study?

8. What is the indication of a more convergent and stronger distribution of CCs in the type of data representations shown in Figure 5?

9. Which differential transformation order is found to be more appropriate according to the strength and convergence of CCs in the three regions of interest?

6. PLOS authors have the option to publish the peer review history of their article (what does this mean?). If published, this will include your full peer review and any attached files.

Reviewer #1: No

Reviewer #2: No

Reviewer #3: **Yes: **Noureddine Elboughdiri

---

## [Author Response · Author response to Decision Letter 0]

5 Apr 2023

Response to reviewers are given in the file attached. All three reviewers' comments/suggestions are incorporated in the letter.

---

## [Decision Letter · Decision Letter 1]

7 May 2023

Fractional order ATR-FTIR Differential Spectroscopy for Detection of Weak Bands and Assessing the Radiation Modifications in Gamma Sterilized UHMWPE

PONE-D-22-32699R1

Dear Dr. Muddassar,

We’re pleased to inform you that your manuscript has been judged scientifically suitable for publication and will be formally accepted for publication once it meets all outstanding technical requirements.

Kind regards,

Hannes C Schniepp, Dr. sc. nat.

Academic Editor

PLOS ONE

Additional Editor Comments (optional):

Reviewers' comments:

Reviewer's Responses to Questions

**Comments to the Author**

1. If the authors have adequately addressed your comments raised in a previous round of review and you feel that this manuscript is now acceptable for publication, you may indicate that here to bypass the “Comments to the Author” section, enter your conflict of interest statement in the “Confidential to Editor” section, and submit your "Accept" recommendation.

Reviewer #3: All comments have been addressed

2. Is the manuscript technically sound, and do the data support the conclusions?

Reviewer #3: Yes

3. Has the statistical analysis been performed appropriately and rigorously? 

Reviewer #3: Yes

4. Have the authors made all data underlying the findings in their manuscript fully available?

Reviewer #3: Yes

5. Is the manuscript presented in an intelligible fashion and written in standard English?

Reviewer #3: Yes

6. Review Comments to the Author

Reviewer #3: Accept in the current form

Accept in the current form

Accept in the current form

Accept in the current form

Accept in the current form

7. PLOS authors have the option to publish the peer review history of their article (what does this mean?). If published, this will include your full peer review and any attached files.

Reviewer #3: **Yes: **Noureddine Elboughdiri

---

## [Editor Report · Acceptance letter]

16 May 2023

PONE-D-22-32699R1 

Fractional order ATR-FTIR differential spectroscopy for detection of weak bands and assessing the radiation modifications in gamma sterilized UHMWPE 

Dear Dr. Muddassar:

I'm pleased to inform you that your manuscript has been deemed suitable for publication in PLOS ONE. Congratulations! Your manuscript is now with our production department. 

Kind regards, 

on behalf of

Dr. Hannes C Schniepp 

Academic Editor

PLOS ONE